# Conspiracy Beliefs Are Associated with Lower Knowledge and Higher Anxiety Levels Regarding COVID-19 among Students at the University of Jordan

**DOI:** 10.3390/ijerph17144915

**Published:** 2020-07-08

**Authors:** Malik Sallam, Deema Dababseh, Alaa’ Yaseen, Ayat Al-Haidar, Nidaa A. Ababneh, Faris G. Bakri, Azmi Mahafzah

**Affiliations:** 1Department of Pathology, Microbiology and Forensic Medicine, School of Medicine, the University of Jordan, Amman 11942, Jordan; alaa_mohamad1990@yahoo.com (A.Y.); mahafzaa@gmail.com (A.M.); 2Department of Clinical Laboratories and Forensic Medicine, Jordan University Hospital, Amman 11942, Jordan; 3Department of Translational Medicine, Faculty of Medicine, Lund University, 22184 Malmö, Sweden; 4School of Dentistry, the University of Jordan, Amman 11942, Jordan; deemahameddababseh@gmail.com (D.D.); ayat_alhaidar@yahoo.com (A.A.-H.); 5Cell Therapy Center (CTC), the University of Jordan, Amman 11942, Jordan; nidaaanwar@gmail.com; 6Department Internal Medicine, School of Medicine, the University of Jordan, Amman 11942, Jordan; fbakri@yahoo.com; 7Department of Internal Medicine, Jordan University Hospital, Amman 11942, Jordan; 8Infectious Diseases and Vaccine Center, University of Jordan, Amman 11942, Jordan

**Keywords:** novel coronavirus, SARS-CoV-2, Middle East, Facebook, Instagram, Twitter, WhatsApp, TV, news

## Abstract

The world has been afflicted heavily by the burden of coronavirus disease 2019 (COVID-19) that overwhelmed health care systems and caused severe economic and educational deficits, in addition to anxiety among the public. The main aim of this study was to evaluate the mutual effects of belief that the pandemic was the result of a conspiracy on knowledge and anxiety levels among students at the University of Jordan (UJ). An electronic-based survey was conducted between 29 March, 2020 and 31 March, 2020. The targeted population involved all undergraduate and postgraduate students from the health, scientific and humanities schools at UJ. Survey sections included 26 items on: socio-demographic information, knowledge and sources of information about the disease, attitude towards the false notion that COVID-19 stemmed from a conspiracy and items to assess the anxiety level among students during the quarantine period. The total number of participants was 1540 students. The mean age of study participants was 22 years and females predominated the study population (n = 1145, 74.4%). The majority of participants perceived the disease as moderately dangerous (n = 1079, 70.1%). Males, Jordanians and participants with lower income were more inclined to feel that COVID-19 is very dangerous. A lower level of knowledge and a higher level of anxiety about COVID-19 were associated with the belief that the disease is part of a conspiracy. Females and participants with lower income were more likely to believe that the disease is related to conspiracy. Belief in conspiracy regarding the origin of COVID-19 was associated with misinformation about the availability of a vaccine and the therapeutic use of antibiotics for COVID-19 treatment. The Ministry of Health in Jordan was the most common source of information about COVID-19 reported by the participants (n = 1018). The false belief that COVID-19 was the result of a global conspiracy could be the consequence of a lower level of knowledge about the virus and could lead to a higher level of anxiety, which should be considered in the awareness tools of various media platforms about the current pandemic.

## 1. Introduction

Humankind is under continuous threat elicited by emerging and re-emerging infectious diseases and the current coronavirus disease 2019 (COVID-19) pandemic is the full-blown manifestation of such a threat [1,2,3]. In 2020, the World Health Organization (WHO) declared the disease caused by severe acute respiratory syndrome coronavirus 2 (SARS-CoV-2) as a pandemic [4]. The world was overwhelmed by the rapid escalation of events, exponential increase in the number of cases and mortality rate of the disease, which was reported in China, the first epicenter of COVID-19 [5,6,7].

The novel respiratory disease COVID-19, has a median incubation period of five days (2–14 days) with the most common symptoms including fever, dry cough and fatigue [8]. Other signs and symptoms that were reported to a lesser degree included productive cough, dyspnea, myalgia, sore throat and headache [9].

With no specific antiviral treatment options available so far, the prevention of the disease remains the mainstay approach to halt the spread of the virus [10]. The preventive measures revolve mainly around social distancing and strict quarantine [11]. Avoiding crowded places and keeping a safe distance from anyone are considered among the most important preventive measures, as SARS-CoV-2 is known to be transmitted via droplets [12,13]. Since the virus is also known to be transmitted through close contact, any form of physical social greetings should be avoided [12]. The preventative approach also includes: practicing regular hygiene and sanitation measures such as hand washing, constant disinfection of surfaces and wearing masks and gloves as appropriate [14]. Abstaining from travel and avoiding people who have been to countries highly impacted by the pandemic is of paramount importance [14]. It is crucial as well to prevent the spread of infection, which is done by practicing coughing and sneezing etiquette and self-isolating in case of suspected infection with the virus [11,14].

Despite the rapid increase in the number of publications regarding COVID-19 in the literature, some aspects of the disease have not been clearly identified yet. This vagueness can lead to a huge stream of misinformation about the virus and the disease [15]. These aspects involve the origin of the virus, and availability of a specific antiviral treatment and effective vaccine, in addition to questioning the accuracy of the recently developed diagnostic modalities [15]. In the current day and age, the widespread access to the internet and the extensive use of social media outlets to get information can be a double-edged sword [16,17,18]. On one end, information can be delivered to a huge target population within a short period of time. However, this information might be faulty and can spread easily without having a credible source [16,18,19]. 

The fear and anxiety that accompanied the COVID-19 pandemic can have devastating effects on the mental health of people and might have a negative psychological and social impact [20,21]. Providing accurate and timely information can result in the clearance of vagueness and relief of anxiety [19]. 

Conspiracy beliefs can be defined as unsubstantiated and implausible beliefs that involve the role of a malevolent force in plotting major events, when other explanations are more probable [22]. Such beliefs can have negative health and social effects which were seen in the past and continue to exist to this day [23]. The striking example regarding the effect of conspiracy is the HIV epidemic in South Africa, where the belief in conspiracy resulted in governmental policies with devastating effects on public health [24]. Another example, is the vaccination conspiracy theories, with sinister outcomes manifested by the re-occurrence of outbreaks of infectious diseases including measles, mumps and rubella [25,26]. 

Jordan was affected by COVID-19 similar to most countries of the world, with 381 reported cases and seven deaths as a result of the disease, at the time of manuscript writing [7]. On 18 March, curfew and quarantine were first implemented in the country following earlier closure of schools and universities with recommendations to avoid large gatherings, which took place on 15 March. The students in general and particularly university students were heavily affected by the conditions surrounding the disease including forced stay at home during the quarantine. Thus, the aims of the current study were to evaluate the overall knowledge about COVID-19 and SARS-CoV-2 among the University of Jordan (UJ) students from different schools. In addition, we aimed to assess the attitude of UJ students towards the perceived danger of the disease and the quarantine measures issued by the government. Moreover, we sought to evaluate possible deleterious effects of conspiracy belief regarding the origin of COVID-19, particularly the potential impact on knowledge and anxiety levels among UJ students. Finally, we aimed to investigate the sources of information about the disease among the students. 

## 2. Materials and Methods

This study was conducted using an online-based questionnaire that was distributed among students at UJ, which is the largest and oldest university in Jordan with about 49,000 enrolled students as of the 2019/2020 academic year [27]. Despite its caveats, an online-based survey was the sole sampling strategy that was feasible considering the conditions of lockdown in Jordan during sampling. Participation in the study was voluntary and an informed consent was provided at the introductory section of the questionnaire (Appendix A).

### 2.1. Description of the Questionnaire

Pilot testing involved distributing the questionnaire draft to seven participants (who did not take part in the final survey), which resulted in minor changes in language and content. The questionnaire was distributed in Arabic and comprised 26 items on socio-demographic information (age, nationality, gender, university program and school, marital status and monthly income of the family), conditions of living during the quarantine period and attitude regarding the perceived danger of the disease and towards the adherence to quarantine measures. To assess knowledge of the participants on the virus and the disease, several items were included that evaluated signs and symptoms, transmission routes, protective measures, therapeutic use of antibiotics, availability of a vaccine for COVID-19 and belief in the assumption that summer heat can inactivate SARS-CoV-2. In addition, the survey items included a section on sources of information about the disease including social media platforms. A specific item was used to evaluate the belief of each participant that the COVID-19 pandemic is part of a global conspiracy. Finally, a section to assess the anxiety level among participants was included and comprised seven questions with four potential responses.

Invitation to participate in this survey was distributed among UJ students via Facebook and WhatsApp. The survey was conducted from 29 March 2020 (14:00) till 31 March (16:00), thus spanning 50 h.

### 2.2. Ethical Permission

The study was approved by the Department of Pathology, Microbiology and Forensic Medicine and by the Scientific Research Committee at the School of Medicine/UJ (Meeting #2 of week 14, 2020 using WhatsApp conference call). In addition, the study was approved by the Institutional Review Board (IRB) at Jordan University Hospital (Ref. No. 10-2020-8556, decision 80/2020). Participation in the study was voluntary and anonymous. An informed consent was ensured by the presence of an introductory section of the questionnaire, with submission of responses implying the agreement to participate. All collected data were treated confidentially.

### 2.3. COVID-19 Knowledge Score (K-Score) Calculation

To evaluate the overall knowledge of each participant about COVID-19 and SARS-CoV-2, a total of 12 items each worth of a single point were included with a correct response to each item being considered as a single point yielding a maximum score of 12. These items involved questions on signs and symptoms (five items with one point weight for each correct answer), routes of transmission (four items with one point awarded for each correct answer, and for blood transmission, non-selection was regarded as a correct answer), antibiotic treatment, availability of a vaccine and effect of summer heat on the epidemic, each worth a single point for each correct answer. 

### 2.4. Anxiety Score Calculation

We based the anxiety score system on the 7-item Generalized Anxiety Disorder Scale (GAD-7), which is a reliable and commonly used system to assess the level of anxiety [28]. This system is modelled based on four possible responses (“not at all response” was scored as zero, “several days” response was scored as 1, “more than half the days” was scored as 2 and “everyday” was scored as 3) to seven questions about their feelings during the past two weeks prior to survey in our study (the quarantine period, Appendix A) [29]. The maximum score of 21 was regarded as the highest level of anxiety, while zero was considered to represent the lowest level. A GAD-7 scale of 0–4 indicates no anxiety, 5–9 indicates mild anxiety, 10–14 indicates moderate anxiety and a score of 15 and above indicates severe anxiety [29].

### 2.5. Statistical Analysis

We used a chi-squared test (χ^2^) to examine the significance of relationships between categorical variables. To compare differences between two independent groups when the dependent variable is continuous, we used the Mann–Whitney *U* test (M–W), and for more than two independent groups, we used the Kruskal–Wallis test (K–W) instead. We also used a two-sided *t*-test to compare differences between the means of two groups. *p*-values less than 0.050 were considered significant. All statistical analyses were conducted in IBM SPSS Statistics 22.0 for Windows (IBM, Armonk, NY, U.S.). 

## 3. Results

### 3.1. Characteristics of the Study Population

The total number of participants in the study who completed the questionnaire was 1540. The mean age of study participants was 22 years (median: 21 years, interquartile range (IQR): 20–22 years). Females predominated the study population (n = 1145, 74.4%) and the majority were Jordanians (n = 1386, 90.2%). Undergraduate students comprised 89.5% (n = 1378) of the study participants and 43.1% (n = 664) were students at health schools, with the highest participation from the School of Dentistry (n = 259, 16.8%), while the lowest number was from the School of Law (n = 4, 0.3%, Appendix A). The highest number of study participants reported a household monthly income of JOD 500–1000 (n = 646, 41.9%). The vast majority of participants were single (n = 1440, 94.1%, Table 1) and spent the last two weeks of curfew with their families (n = 1407, 91.7%).

### 3.2. Knowledge of COVID-19 Transmission, Prevention and Control

Regarding knowledge on signs and symptoms of the disease, fever was the most frequent sign to be correctly identified by the participants (n = 1500, 97.4%) followed by shortness of breath (n = 1448, 94.0%) and cough (n = 1309, 85.0%).

For possible transmission routes of the virus, touching infected surfaces (fomites) was the most common route to be correctly identified (n = 1485, 96.4%) followed by coughing and sneezing (n = 1325, 86.0%). Close contact in crowded places was missed as a potential route of transmission in 21.0% of the participants (n = 324). Transmission via blood was incorrectly identified by 17.8% of the participants (n = 274).

Of the eight protective and control measures that were asked in the survey, the majority of participants precisely identified all protective measure (n = 1193, 77.5%) and an additional 134 participants missed only a single protective measure out of the eight items in the survey (8.7%), followed by 85 participants who missed two items (5.5%).

Regarding the current lack of an effective vaccine against SARS-CoV-2, the vast majority of participants provided a correct answer (n = 1433, 93.1%). In addition, the majority of participants identified the useless effect of antibiotics in treating COVID-19 (n = 1365, 88.6%). Summer heat ability to inactivate the virus was incorrectly reported by 40.3% of the study participants (n = 621).

### 3.3. Attitude Towards COVID-19

Regarding the attitude of the participants towards the perceived danger of the disease, the majority reported that COVID-19 is moderately dangerous (n = 1079, 70.1%) and 428 participants reported that the disease is very dangerous (27.8%). Males were more likely to report that COVID-19 is very dangerous compared with females (38.6% vs. 24.1%, *p* < 0.001; χ^2^, Table 2). Jordanian participants had a significantly higher likelihood to report that the disease is very dangerous compared with their non-Jordanian colleagues (28.5% vs. 21.2%, *p* = 0.020; χ^2^, Table 2). Participants with families of lower income were more likely to feel that COVID-19 is very dangerous compared with those with higher income (38.0% vs. 22.1%, *p* < 0.001; χ^2^, Table 2). A higher level of anxiety was found more frequently among participants who felt the disease is more dangerous as estimated using anxiety scores (mean anxiety score of 8.1 among those who reported the disease as moderately dangerous, as opposed to mean anxiety score of 9.2 among those who reported that COVID-19 is very dangerous, *p* < 0.001; K–W, Figure 1). Correlation of age, level of study, marital status and belief in conspiracy regarding the origin of the virus with perception of COVID-19 danger did not result in statistically significant differences (Table 2). The vast majority of participants followed the government-issued quarantine measures (n = 1506, 98.2%). Married participants were less likely to adhere to quarantine measures compared with single students (7.1% vs. 1.5%, *p* = 0.001; χ^2^). In addition, male participants were less likely to adhere to quarantine measures compared with females (3.3% vs. 1.2%, *p* = 0.007; χ^2^). Further, postgraduate students were more likely to break the quarantine measures (3.7% vs. 1.5%, *p* = 0.047; χ^2^).

### 3.4. Correlation of COVID-19 Knowledge to Different Variables

The overall knowledge regarding the disease and the virus among the participants was generally high with 9.5 as the mean K-score. Older age was associated with a higher level of knowledge (mean K-score: 9.7 vs. 9.3, *p* < 0.001; *t*-test). Postgraduate students had higher mean K-scores compared with their undergraduate counterparts (9.7 vs. 9.5, *p* = 0.035; M–W). The highest mean K-score was found among students at health schools followed by the scientific schools, while the lowest mean K-score was found among the humanities schools (9.8 vs. 9.4 vs. 9.1, *p* < 0.001; K–W). A higher mean K-score was observed among those who felt that COVID-19 is very dangerous compared with those who felt that the disease is moderately dangerous, however, without statistical significance (*p* = 0.150; K–W). For gender, marital status, nationality and family income, no statistically significant differences were found as well.

The level of knowledge about COVID-19 was lower among participants who believed that the disease is part of a conspiracy, compared with those who did not have such a belief (mean K-score: 9.0 vs. 9.7, *p* < 0.001; M–W). In addition, the mean K-score showed a gradual decrease going from those who denied the existence of conspiracy to those who answered maybe, and ending in participants who had such a belief (mean K-score: 9.0 vs. 9.5 vs. 9.7; *p* < 0.001; K–W, Figure 2).

### 3.5. Anxiety Level in Relation to Other Variables

For the whole study population, the mean anxiety score was 8.4 (median = 8.0, IQR (5.0–12.0)). Males showed a lower level of anxiety compared with females (mean anxiety scores: 7.7 vs. 8.6, *p* = 0.002; M–W). A higher level of anxiety was found among participants with the lowest monthly income as compared with the other two groups (8.9 vs. 8.3 vs. 8.1, *p* = 0.043; K–W, Appendix A). In addition, a higher level of anxiety was also noticed among non-Jordanians, however, without statistical significance (mean anxiety score: 9.2 vs. 8.3, *p* = 0.068; M–W). Nevertheless, a significantly higher level of anxiety was found among non-Jordanian females compared with their Jordanian counterparts (mean anxiety score: 10.0 vs. 8.5; *p* = 0.011; M–W, Appendix A).

Anxiety scores were significantly higher among the study participants who believed that COVID-19 is the result of a global conspiracy compared with those who denied such a belief (mean anxiety score: 9.0 vs. 7.7, *p* = 0.004; M–W). In addition, a gradual increase in the level of anxiety was observed moving from a mean score of 7.7 among those who did not believe in the role of conspiracy, to a mean score of 8.6 among those who answered maybe, and reaching the highest mean score of 9.0 among those who had such a belief (*p* = 0.001; K–W, Figure 3). On the other hand, no significant differences in anxiety scores were observed for study level, schools, nationalities and conditions of living during the quarantine.

### 3.6. Association of Belief in Conspiracy with Other Variables

The total number of participants who stated that COVID-19 is not part of a conspiracy was 518, representing 33.6% of the study population. On the other hand, 16.4% of the participants stated that they believe in the role of conspiracy in the origin of the disease (n = 253), and those who answered maybe represented 49.9% of the study population (n = 769).

Upon comparing different variables depending on whether the participants had a belief that COVID-19 is part of a global conspiracy or not, we found that females were more inclined to have such a belief compared with males (36.3% vs. 23.8%; *p* = 0.001; χ^2^ test). Those with lower income had a higher likelihood to believe that the disease is the result of a global conspiracy (74.6% answered yes or maybe among participants with an income of JOD < 500, compared with 59.0% who answered yes or maybe among participants with an income of JOD > 1000, *p* < 0.001; χ^2^ test). In addition, the participants who believed in conspiracy had a higher tendency to believe that there is a vaccine for COVID-19 (8.2% vs. 4.4%, *p* = 0.001; χ^2^ test), and to believe that the disease can be treated with antibiotics (13.7% vs. 6.8%, *p* < 0.001; χ^2^ test). Further, those who believed either entirely or indicated a possibility of conspiracy had the doubtful belief that summer heat will inactivate SARS-CoV-2 (45.8% vs. 29.5%, *p* < 0.001; χ^2^ test). Level of study, nationality and living conditions during the quarantine did not have statistically significant differences in relation to belief in conspiracy.

### 3.7. Sources of Information Regarding COVID-19

Regarding the most frequent sources of information about the disease and the virus that were reported by the students, the Jordanian Ministry of Health (MoH) website was the most frequent one (n = 1018), followed by television programs and news releases (n = 918), social media (n = 913), medical doctors (n = 684), scientific journals (n = 462) and the UJ websites (n = 362). The WHO website was the most frequently reported source of information mentioned in the “others” option (n = 18). The majority of students reported more than one source of information about the pandemic (n = 1238). For social media platforms, Facebook was the most frequent source of information (n = 911), followed by Instagram (n = 283), WhatsApp (n = 270) and Twitter (n = 209). 

## 4. Discussion

Knowledge and attitude surveys can be used as an asset to identify gaps in knowledge, and certain misbeliefs and patterns of behavior. Such gathered information can be helpful to plan for better action, especially in the current time where the public appears vulnerable to media littered with piles of misinformation that lacks accuracy at times [16]. This misinformation can create a response that varies from full-fledged terror and panic to complete negligence, which can impede a successful response to the current pandemic in both ways [15,19]. The response to COVID-19 requires cooperation of the public through following government-issued strict quarantine and social distancing measures, which are the best strategies to lessen the effects of the pandemic and to prevent the collapse of health care systems [14]. This also applies to university students where the conditions that accompanied the pandemic have resulted in psychological and educational difficulties [30]. The swift spread of COVID-19 pandemic was met initially by alertness of the public in some countries, but confusion and panic in others. Panic could particularly be related to the lack of accurate information about the pandemic [4,31,32]. 

One of the main misconceptions is the presence of conspiracy theories regarding the origin of the virus itself [15,33]. The interplay between virus evolution and dynamics of virus emergence, diversification and spread has been reviewed by Pybus and Rambaut [34]. Virus evolution by itself is not the sole factor for the increased incidence of infectious disease. Increased human mobility with ease of travel and growth of the global population resulted in increased contact with virus reservoirs and vectors for transmission of human infectious agents and rapid global spread of these novel pathogens [35]. Obvious examples to illustrate this phenomenon include the spread of West Nile fever from the Middle East into New York in 1999 and the emergence of severe acute respiratory syndrome (SARS) coronavirus in 2003, Middle East respiratory syndrome coronavirus (MERS) in 2012, Ebola in 2014 and Zika fever in 2015 [36,37,38,39]. Thus, conspiracy theories regarding origins of viral diseases, including COVID-19 are not plausible on any scientific level. Currently, there is sufficient conclusive evidence that explains the origin of SARS-CoV-2 from a bat reservoir [40,41]. Other currently circulating misconceptions include the presence of an effective treatment for COVID-19 using antibiotics (azithromycin) and antimalarials (hydroxychloroquine) that have not been proven as effective treatment yet, with variable and conflicting results [42]. In addition, there is a widespread belief that the pandemic will die out in summer, despite the absence of a clear-cut evidence of such a notion. On the contrary, the spread of SARS-CoV-2 in the Southern Hemisphere might hint to the fallacy of such a claim. Thus, the discernable effect of summer heat on the virus needs further scientific investigation to reach valid and conclusive evidence about this issue.

The main study findings are the following: students at the UJ displayed a relatively high level of knowledge about COVID-19, which was shown by high mean K-scores. More than 80% of the participants correctly identified the commonly reported signs and symptoms of COVID-19 (fever, cough and shortness of breath). In addition, the students showed a high level of knowledge regarding the transmission routes (fomites and droplet transmission through sneezing or coughing) and the preventive methods. This level of knowledge might be attributed to mass awareness campaigns via different media channels including the UJ, MoH and news websites and their social media pages and accounts, besides the massive awareness campaigns on TV and the internet. Further, this result can stem from the desire of students to actively seek knowledge about this disease that strongly affected their lives including educational, social and mental aspects. However, important gaps in knowledge regarding other possible modes of transmission were identified. This included about one-fifth of the participants missing the importance of crowded places as a possible setting for virus transmission. Such places can increase the chance of exposure to respiratory droplets, with the possibility of transmission from infected people lacking symptoms [13,43,44]. In addition, transmission via blood was incorrectly identified by 17.8% of the students. Despite concerns related to infrequent detection of SARS-CoV-2 in blood, no evidence of confirmed or even suspected blood-borne cases of COVID-19 were reported. This pattern was also seen in SARS and MERS, the two other recent emerging coronavirus epidemics [45,46]. In addition, the participants had sufficient knowledge regarding the unavailability of an effective vaccine and the uselessness of antibiotics for COVID-19.

Regarding the attitude towards COVID-19 danger, males were more likely to perceive the disease as very dangerous compared with females. This perception can be ascribed to the financial and economic by-products of the pandemic and the quarantine, and their fear of what is at stake from the mandatory unemployment. Similarly, lower monthly family income was associated with a higher perception of danger. As expected, participants who perceived the virus as very dangerous had higher levels of anxiety. 

The vast majority of participants showed positive attitude towards the quarantine through following the government-issued rules. However, a minority (1.8%) stated that they broke the quarantine. Those were more likely to be males, married and postgraduate participants, and a possible explanation for their attitude is that they are the ones who venture outside to buy groceries or in case of emergencies. 

Participants from health schools had higher COVID-19 knowledge, compared with participants from the science and humanities schools, which might be related to the possibility of having similar subjects in their curriculum, and their general understanding of diseases. Moreover, the mean K-score was also higher among the older and the postgraduate students, which seems plausible.

Based on the previous explanation in the Methods section, the results indicated that the study population had mild anxiety with an 8.4 mean anxiety score. Keeping in mind that a score of 10 and above warrants further psychological assessment and in some cases therapy, the UJ is advised to take an active reassuring approach towards the students, together with providing accurate and timely information about the disease. In addition, females had higher anxiety levels: females tend to worry and overthink more, leading to anxiety, as opposed to males who use distractions as a coping mechanism [47,48]. Additionally, non-Jordanian females had significantly higher anxiety levels, and this can be the result of being abroad and probably spending the quarantine away from their families, causing them to be more anxious.

Participants with lower monthly income had a higher anxiety level. This can be partially explained by the fact that the aforementioned group mostly depends on day-to-day income, with obvious financial impediments during the quarantine period.

Regarding the overall belief in the conspiracy in relation to the current pandemic, only a third of the participants rejected such a claim, whereas the majority of the students either believed entirely or at least had an inclination to believe in this dubious and even harmful notion. This harmful way of thinking might have negative consequences on people’s psychological, social and health status. Examples of these negative impacts may include the possibility of racial abuse through a distrustful view of other people and anti-vaccination campaigns [49].

In general, those who believed in the conspiracy and even those who were skeptical about it had lower knowledge about SARS-CoV-2. A tangible explanation for the belief in these conspiracies is the lack of proper knowledge about the disease. The results of the study also showed a clear relation between the belief in the conspiracies and the elevated levels of anxiety. 

The majority of students who believed in the conspiracy were female participants and those with lower income. Participants who believed in the conspiracy were also more likely to believe that a vaccine is available and the disease can be treated using antibiotics. They also thought that summer heat is capable of inactivating the virus. 

Finally, our results showed that the main sources of information for the students were the MoH website on COVID-19, TV, news releases and Facebook [50]. Thus, these media outlets should take a meticulous approach in rigorously reviewing the accuracy of information they provide about the disease, taking into account the reliance of the public in general and students in particular on these sources to get knowledge about the current pandemic. 

This study had several important limitations. First, an inevitable caveat in all surveys is the tendency of some participants to respond in a way they believe to be suitable for the researchers. Second, during the sharing of the survey, it was emphasized to answer to the best of participants’ knowledge, however, there is never a guarantee that they followed such an instruction. Furthermore, willingness to participate, especially on an online-based survey, may have been limited. Other shortcomings were the female predominance in the sample and higher number of participants that were affiliated with health and scientific schools. One important point should be clarified, which is related to the scoring system that was used to assess the level of knowledge among participants regarding COVID-19. This assessment tool might be arbitrary and subjective. In addition, the weight of each item can be criticized considering the difficulty in assessing the contribution of each item to overall COVID-19 knowledge. Thus, generalizability of our results in relation to this issue should be made with extreme caution.

## 5. Conclusions

The impact of the COVID-19 pandemic is not merely related to health issues, but also involves social and psychological effects. The results of this study highlight the negative effect of misinformation that is conveyed by media teeming with fallacies and assumptions that lack substantial evidence, particularly the belief in a conspiracy role in the pandemic. The negative impact on UJ students was revealed by a significantly higher level of anxiety and lower knowledge about COVID-19 in those who believed in these claims of conspiracy. This must be addressed by the main sources of knowledge that were identified by the participants (e.g., MoH, TV, social media outlets), which are encouraged to have robust fact-checking processes before conveying information about this unprecedented pandemic.

This study identified gaps in the knowledge among UJ students about COVID-19, particularly among those studying at humanities schools. Thus, it is crucial to sustain and intensify the awareness and education of students with evidence-based knowledge. 

## Figures and Tables

**Figure 1 ijerph-17-04915-f001:**
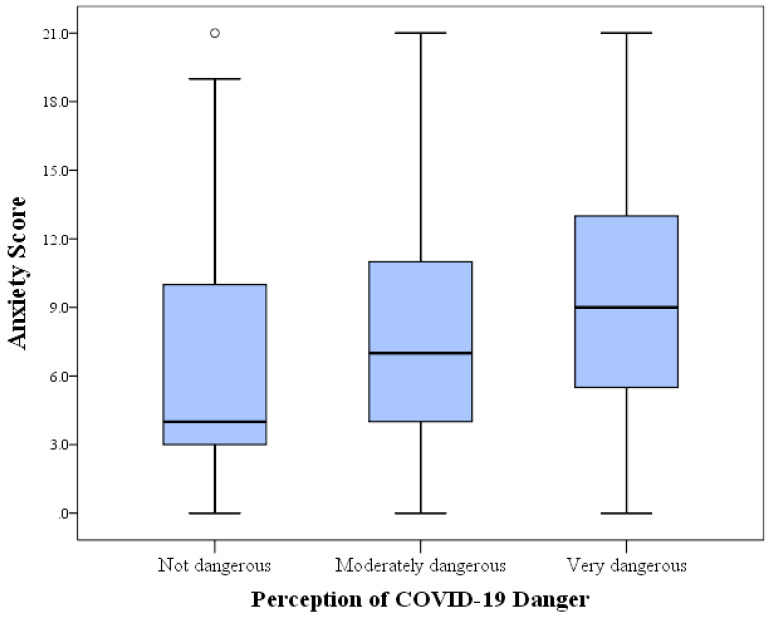
Anxiety score distribution among the study participants stratified by attitude towards COVID-19 perceived danger. Gradual increase in anxiety was seen among students at the University of Jordan in relation to their perception of coronavirus disease 2019 (COVID-19) danger. The difference was statistically significant (*p* < 0.001, Kruskal–Wallis test).

**Figure 2 ijerph-17-04915-f002:**
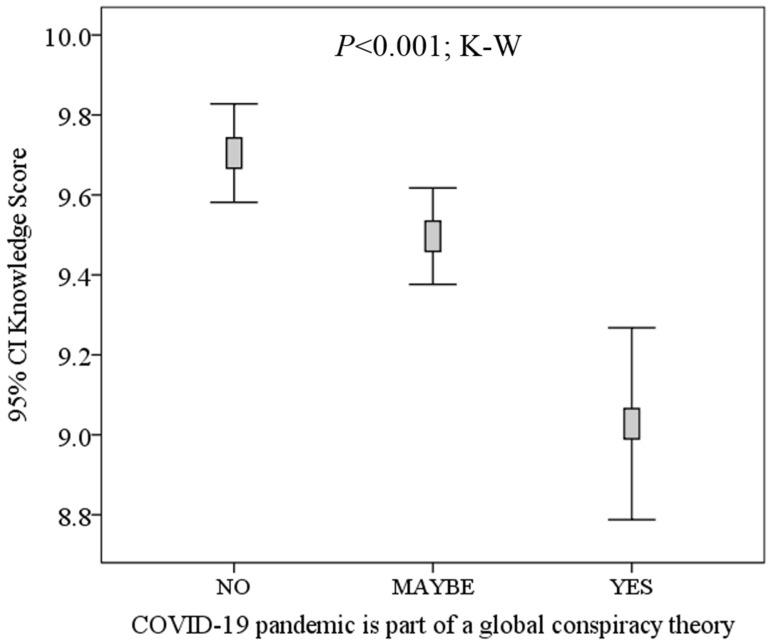
Correlation between students’ knowledge about COVID-19 and their belief in a conspiracy role in the disease. Participants were students at the University of Jordan. K–W: Kruskal–Wallis test. CI: confidence interval.

**Figure 3 ijerph-17-04915-f003:**
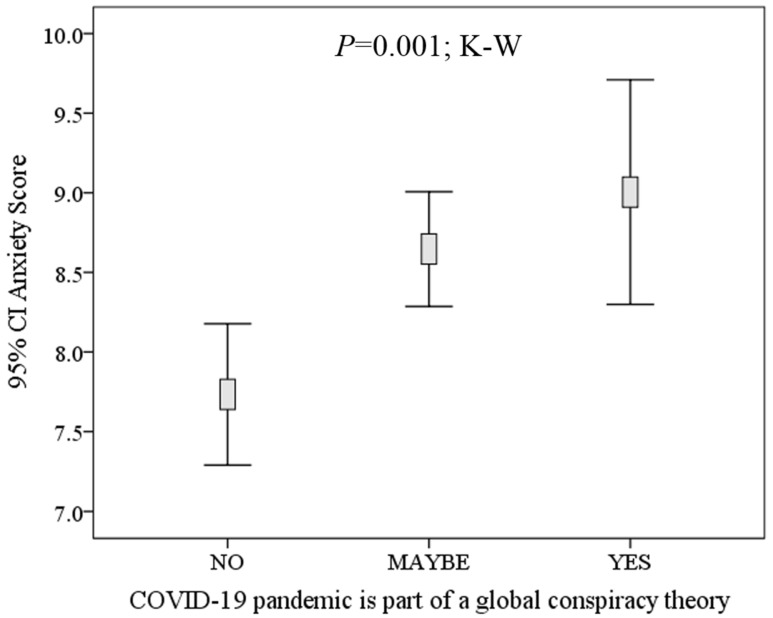
Correlation between students’ anxiety about COVID-19 and their belief in a conspiracy role in the disease. Participants were students at the University of Jordan. K–W: Kruskal–Wallis test. CI: confidence interval.

**Table 1 ijerph-17-04915-t001:** Characteristics of the study participants.

Characteristic	N ^1^ (%)
Age (median, SD ^2^)	21 (3.7)
Gender	
*Male*	394 (25.6)
*Female*	1145 (74.4)
Nationality	
*Jordanian*	1386 (90.2)
*Non-Jordanian* ^3^	151 (9.8)
Program ^4^	
*BSc*	1378 (89.5)
*MSc*	138 (9.0)
*PhD*	24 (1.6)
Schools ^5^	
*Health* ^6^	664 (48.5)
*Scientific* ^7^	392 (28.6)
*Humanities* ^8^	313 (22.9)
Marital status	
*Single*	1440 (94.1)
*Married*	85 (5.6)
*Divorced*	6 (0.4)
Monthly income ^9^	
*Less than JOD 500* ^10^	397 (25.8)
*JOD 500–1000*	646 (41.9)
*More than JOD 1000*	497 (32.3)

^1^ N: number, some categories will not add up to 1540 because of missing information; ^2^ SD: standard deviation; ^3^ non-Jordanian: participants of non-Jordanian origin included 22 different nationalities, with the most common being Palestine (n = 42), Iraq (n = 33) and Kuwait (n = 28); ^4^ program: BSc is Bachelor of Science, MSc is Masters of Science and PhD is Doctor of Philosophy; ^5^ Schools of Business and King Abdullah II School of Information Technology were excluded from this analysis because they represent two different categories (humanities for the former and scientific for the later); ^6^ health schools: include the Schools of Medicine, Dentistry, Pharmacy, Nursing and Rehabilitation Sciences; ^7^ scientific schools: include the Schools of Engineering, Agriculture, and Science; ^8^ humanities: include Schools of Arts and Foreign Languages, Physical Education, Archaeology and Tourism, Sharia, Educational Sciences, Arts and Design and Law; ^9^ monthly income: the self-reported monthly income of the family; ^10^ JOD: Jordanian dinar.

**Table 2 ijerph-17-04915-t002:** Response of study participants regarding danger of COVID-19, knowledge and belief in conspiracy.

Feature		Nationality			Gender			Schools of UJ ^1^			Monthly Income of the Family ^2^			
		Jordanian	Non-Jordanian ^3^		Male	Female		Health and Scientific Schools ^4^	Humanities Schools ^5^		Less than JOD 500 ^6^	JOD 500–1000	More than JOD 1000	
Survey item	***Response***	**N ^7^ (%)**	**N (%)**	*p*-value ^8^	**N (%)**	**N (%)**	*p*-value	**N (%)**	**N (%)**	*p*-value	**N (%)**	**N (%)**	**N (%)**	*p*-value
Is COVID-19 dangerous? ^9^	***Not dangerous***	26 (1.9)	7 (4.6)	0.020	9 (2.3)	24 (2.1)	<0.001	24 (2.3)	8 (2.6)	<0.001	10 (2.5)	10 (1.5)	13 (2.6)	<0.001
	***Moderately dangerous***	965 (69.6)	112 (74.2)		233 (59.1)	845 (73.8)		782 (74.1)	191 (61.0)		236 (59.4)	469 (72.6)	374 (75.3)	
	***Very dangerous***	395 (28.5)	32 (21.2)		152 (38.6)	276 (24.1)		250 (23.7)	114 (36.4)		151 (38.0)	167 (25.9)	110 (22.1)	
Coronavirus infection can be treated using an antibiotic	***Correct response***	1226 (88.5)	137 (90.7)	0.403	348 (88.3)	1016 (88.7)	0.826	965 (91.4)	249 (79.6)	<0.001	337 (84.9)	577 (89.3)	451 (90.7)	0.018
	***Incorrect response***	160 (11.5)	14 (9.3)		46 (11.7)	129 (11.3)		91 (8.6)	64 (20.4)		60 (15.1)	69 (10.7)	46 (9.3)	
There is a vaccine available for COVID-19	***Correct response***	1288 (92.9)	143 (94.7)	0.414	370 (93.9)	1062 (92.8)	0.436	990 (93.8)	284 (90.7)	0.065	359 (90.4)	602 (93.2)	472 (95.0)	0.029
	***Incorrect response***	98 (7.1)	8 (5.3)		24 (6.1)	83 (7.2)		66 (6.3)	29 (9.3)		38 (9.6)	44 (6.8)	25 (5.0)	
Summer heat can kill the COVID-19 virus	***Correct response***	820 (59.2)	98 (64.9)	0.172	235 (59.6)	684 (59.7)	0.974	701 (66.4)	135 (43.1)	<0.001	198 (49.9)	388 (60.1)	333 (67.0)	<0.001
	***Incorrect response***	566 (40.8)	53 (35.1)		159 (40.4)	461 (40.3)		355 (33.6)	178 (56.9)		199 (50.1)	258 (39.9)	164 (33.0)	
Do you think the COVID-19 pandemic is part of a global conspiracy theory?	***No***	458 (33.0)	58 (38.4)	0.402	163 (41.4)	355 (31.0)	0.001	395 (37.4)	82 (26.2)	<0.001	101 (25.4)	213 (33.0)	204 (41.0)	<0.001
	***Yes***	228 (16.5)	24 (15.9)		51 (12.9)	202 (17.6)		144 (13.6)	69 (22.0)		80 (20.2)	107 (16.6)	66 (13.3)	
	***Maybe***	700 (50.5)	69 (45.7)		180 (45.7)	588 (51.4)		517 (49.0)	162 (51.8)		216 (54.4)	326 (50.5)	227 (45.7)	

^1^ UJ: University of Jordan; ^2^ monthly income: the self-reported monthly income of the family; ^3^ non-Jordanian: participants of non-Jordanian origin included 22 different nationalities, with the most common being Palestine (n = 42), Iraq (n = 33) and Kuwait (n = 28); ^4^ health and scientific schools: include the Schools of Medicine, Dentistry, Pharmacy, Nursing, Rehabilitation Sciences, Engineering, Agriculture and Science; ^5^ humanities schools: include Schools of Arts and Foreign Languages, Physical Education, Archaeology and Tourism, Sharia, Educational Sciences, Arts and Design and Law; ^6^ JOD: Jordanian dinar; ^7^ N: number; ^8^
*p*-value: calculated using chi-squared test (χ^2^); ^9^ COVID-19: coronavirus disease 2019.

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
