# Peer review of "Conspiracy Beliefs Are Associated with Lower Knowledge and Higher Anxiety Levels Regarding COVID-19 among Students at the University of Jordan"

_ijerph, 2020, doi:10.3390/ijerph17144915_

Round 1
Reviewer 1 Report
Dear authors,
Firstly, thanks for submitting your research to the International Journal of Environmental Research and Public Health. Considering the topic addressed in your paper and the value of the contribution done, I believe this paper has, overall, a certain potentiality to be published.
Nevertheless, there are some issues (some of them major, some relatively minor) that should be whether clarified, acknowledged or better-discussed along the paper. Please see my specific comments below, following the same order proposed by you in the text:
- Although the abstract is clear, I would like to suggest the authors to remove the p-values from the results section of it. It results redundant, since usually scientific papers report significant statistical tests, unless otherwise stated.
- On the other hand, the timeframe used for performing the survey is (unlike other studies) adequate and authors did well to clearly explaining it there. However, age mean and sex-based features must be stated there. Also, line 29 is confusing and needs revision.
- The introduction is good (and well-supported), but more articulation of study hypotheses is needed at the end of the section, in order to justify and theoretically support the study aim you state in the last paragraph.
- Also, please provide a bit more of context on the impact of the outbreak in Jordania and the nature and structure of the social measures (e.g. concrete (epidemiological figures, although retrospective; were they locked down?; how did it impact social relationships?).
- Please clearly explain in Materials and Methods the sampling design/strategy, and mention why it was convenient (you did well, in my perspective) and what are the limitations and technical flaws of it for studying this topic.
- The instruments and measures used need a more systematic approach to be described, perhaps integrating these data in a "description of the questionnaire, that includes all the study variables and questionnaire sections" within the methods. Also, please state the reliability measures of the scales there.
- Statistical analysis lacks a better description (very brief) of the nature and purpose for each one of the three principal tests you carried out. Also, it is important to contextualize readers why you did not used ANOVAs an other parametric tests, that for us could be evident, but not necessarily to them.
- Results are fairly presented, although I would suggest re-formatting Table 1, putting the grouping variable in a first (left) column. It will enhance the readability of the table. Also, the footnotes are excessive, and most of these data could be adequately presented in the text.
- The boxplots Figures 1 to 3) do not add a lot of information to the paper. Instead, I'd use simple bar charts with CIs and tendency lines (mean points) between the columns.
- Discussion is good, but authors re-repeat a lot study results, some of them very important but without the adequate external support and other similar studies among all the COVID-19 researches that have been massively published in the last month.
- Further, several issues related to the external validity and test-re-test reliability of the measures should be stated, discussed and clarified by the authors.
- Also limitations should be improved, since authors highly reduced them to technical steps followed in the research, but not, e.g. information sources, personal factors, etc., variables that could act as confounders or moderators.
Best wishes.
Author Response
Dear Editor,
Regarding our manuscript with an ID.: ijerph-811442.
The reviewers’ comments were highly insightful and enabled us to improve the quality of our manuscript. In the following pages are our point-by-point responses to each of the comments of the reviewers.
Please find attached a revised version of our manuscript “Conspiracy beliefs are associated with lower knowledge and higher anxiety levels regarding COVID-19 among students at the University of Jordan”. The revisions were highlighted using the "Track Changes" function in the manuscript file.
We hope that the revisions in the manuscript and our accompanying responses will be sufficient to make our manuscript suitable for publication
Sincerely, and on behalf of the co-authors
Malik Sallam. MD, PhD
Responses to the comments of Reviewer #1
Although the abstract is clear, I would like to suggest the authors to remove the p-values from the results section of it. It results redundant, since usually scientific papers report significant statistical tests, unless otherwise stated.
Response: Based on the reviewer’s suggestion, we omitted the p-values from the abstract.
On the other hand, the timeframe used for performing the survey is (unlike other studies) adequate and authors did well to clearly explaining it there. However, age mean and sex-based features must be stated there. Also, line 29 is confusing and needs revision.
Response: Based on the reviewer’s suggestion, we added the mean age and gender distribution to the abstract as follows: “The mean age of study participants was 22 years and females predominated the study population (n=1145, 74.4%)”. In addition, we rephrased line 29 as follows: “The majority of participants perceived the disease as moderately dangerous (n=1079, 70.1%). Males, Jordanians and participants with lower income were more inclined to feel that the COVID-19 is very dangerous.”
Also, please provide a bit more of context on the impact of the outbreak in Jordania and the nature and structure of the social measures (e.g. concrete (epidemiological figures, although retrospective; were they locked down?; how did it impact social relationships?).
Response: We believe that the study aims were clearly presented at the last paragraph of the introduction section and any further additions will make the text redundant. Thus, we prefer to keep this part of the manuscript in its original form.
The introduction is good (and well-supported), but more articulation of study hypotheses is needed at the end of the section, in order to justify and theoretically support the study aim you state in the last paragraph.
Response: The required info has already been mentioned in the introduction section as follows: “Jordan was affected by COVID-19 similar to most countries of the world, with 381 reported cases and seven deaths as a result of the disease, at the time of manuscript writing. On March 18th, curfew and quarantine were first implemented in the country following earlier closure of schools and universities with recommendations to avoid large gatherings, which took place on March 15th.”
Please clearly explain in Materials and Methods the sampling design/strategy, and mention why it was convenient (you did well, in my perspective) and what are the limitations and technical flaws of it for studying this topic.
Response: Based on the reviewer’s comments we added the following paragraph to the Materials and Methods section: “Despite its caveats, an online-based survey was the sole sampling strategy that was feasible considering the conditions of lockdown in Jordan during sampling.”
The other limitations and technical flaws were fully discussed in the discussion section of the manuscript.
The instruments and measures used need a more systematic approach to be described, perhaps integrating these data in a "description of the questionnaire, that includes all the study variables and questionnaire sections" within the methods. Also, please state the reliability measures of the scales there.
Response: Based on the reviewer’s comment we added a new section to the Materials and Methods section under the subtitle “Description of the questionnaire”. For the reliability of the knowledge and anxiety scores, the robustness of the anxiety scale was previously discussed by Lowe et al, 2008 which was cited in the manuscript. For the knowledge score, we elaborated on its caveats in the discussion section of the manuscript.
Statistical analysis lacks a better description (very brief) of the nature and purpose for each one of the three principal tests you carried out. Also, it is important to contextualize readers why you did not used ANOVAs an other parametric tests, that for us could be evident, but not necessarily to them.
Response: Based on the reviewer’s comment we elaborated more on the rationale behind using each statistical test. For ANOVA, we tried it and it gave the same results.
Results are fairly presented, although I would suggest re-formatting Table 1, putting the grouping variable in a first (left) column. It will enhance the readability of the table. Also, the footnotes are excessive, and most of these data could be adequately presented in the text.
Response: We would like to thank the reviewer for this comment; however, we believe that the table in the current format helps the readers to understand how we analyzed the data. In addition, the footnote included helps the table to stand alone, independent of the text.
The boxplots Figures 1 to 3) do not add a lot of information to the paper. Instead, I'd use simple bar charts with CIs and tendency lines (mean points) between the columns.
Response: We would like to thank the reviewer for this comment; however, we feel that the current figures provide a great visual aid to illustrate the main results of this study. To be more specific, for Figures 2 and 3, there is no crossing between the groups which emphasizes the statistical significance of our results.
Discussion is good, but authors re-repeat a lot study results, some of them very important but without the adequate external support and other similar studies among all the COVID-19 researches that have been massively published in the last month.
Response: We believe that we included the relevant references particularly in relation to the effect of misinformation and conspiracy beliefs.
Further, several issues related to the external validity and test-re-test reliability of the measures should be stated, discussed and clarified by the authors.
Response: We are not familiar with such kind of testing in cross-sectional surveys that test the knowledge, attitude and practices. Thus, we cannot elaborate on this comment.
Also limitations should be improved, since authors highly reduced them to technical steps followed in the research, but not, e.g. information sources, personal factors, etc., variables that could act as confounders or moderators.
Response: We believe that the limitations were thoroughly discussed in the discussion section and we can’t elaborate more.

Reviewer 2 Report
The goal of this manuscript is to investigate the pairwise relationships between COVID-19 knowledge, anxiety, attitudes toward COVID-19, belief in COVID-19 conspiracy and socio-demographic variables among UJ students. The goal is very important to identify young Jordanian adults’ cognitive and emotional responses to COVID-19 that lead to an increase or a decrease in their mental health and adaptive coping behaviors (e.g., wearing masks). However, I found the manuscript to have some weakness in conceptual development and statistical methods. Of my points below, I consider #’s 1, 2 and 3 to be the most serious.
#1. There is no clear reason why the authors selected UJ students as study population, although the authors made out the sentence of ‘The students in general and particularly university students were heavily affected by the conditions surrounding the disease including forced stay at home during the quarantine.’ (lines 92-94). For example, university students are less likely to be anxious about their virus infection because they are young and healthy. They tend to participate in social activities actively so that they are more likely to be infected with COVID-19 and then infect other people with COVID-19. For this reason, the university students can play a crucial role in spreading COVID-19 pandemic in Jordan. It is very important for us to understand their cognitive (e.g., COVID-19 knowledge) and emotional (e.g., anxiety) responses to COVID-19.
#2. I think that the key variables in the manuscript are (1) COVID-19 knowledge, (2) anxiety, (3) attitudes toward COVID-19, and (4) belief in COVID-19 conspiracy. However, I was not able to find the theoretical grounds for authors’ selecting the four variables as the key variables in their study. For example, each of the four variables may positively or negatively affect people’s mental health and/or adaptive coping behaviors during the COVID-19 pandemic. Of course, I think that the authors should make their point of such argument clear based on previous studies.
#3. I think that UJ students’ behavioral responses to COVID-19 (e.g., wearing masks) should be measured in the study. This is because it is very important for us to investigate how cognitive and emotional responses to COVID-19 may affect (or be related to) behavioral responses to COVID-19 in order to offer practical implications related to preventing the spread of COVID-19 in Jordan. I wondered what the authors think about my thoughts.
#4. I believe that students at health schools are different from students at scientific schools in cognitively responding to COVID-19 (e.g., COVID-19 knowledge) as the virus is strongly associated with human health that is mainly studied by students at health schools but not students at scientific schools. Hence, I think that the school variable can be divided into three categories: health schools, scientific schools and humanities schools. I wondered what the authors think about my thoughts.
#5. I’d like to suggest that (1) discussion of ‘2.2. COVID-19 knowledge calculation’ sub-section is inserted after line 115 and (2) discussion of ‘2.3. Anxiety score calculation’ sub-sections is inserted after line 119, in order to increase readers’ understandability of measures.
I hope these comments are useful in moving your research forward.
Author Response
Responses to the comments of Reviewer #2
#1. There is no clear reason why the authors selected UJ students as study population, although the authors made out the sentence of ‘The students in general and particularly university students were heavily affected by the conditions surrounding the disease including forced stay at home during the quarantine.’ (lines 92-94). For example, university students are less likely to be anxious about their virus infection because they are young and healthy. They tend to participate in social activities actively so that they are more likely to be infected with COVID-19 and then infect other people with COVID-19. For this reason, the university students can play a crucial role in spreading COVID-19 pandemic in Jordan. It is very important for us to understand their cognitive (e.g., COVID-19 knowledge) and emotional (e.g., anxiety) responses to COVID-19.
Response: The rationale behind using University students as the study population as this was the first step of a project that will culminate in a broad study involving public in Jordan. The University of Jordan was selected for logistic purposes as the authors are affiliated to the University of Jordan and an invitation would be more likely to be answered.
#2. I think that the key variables in the manuscript are (1) COVID-19 knowledge, (2) anxiety, (3) attitudes toward COVID-19, and (4) belief in COVID-19 conspiracy. However, I was not able to find the theoretical grounds for authors’ selecting the four variables as the key variables in their study. For example, each of the four variables may positively or negatively affect people’s mental health and/or adaptive coping behaviors during the COVID-19 pandemic. Of course, I think that the authors should make their point of such argument clear based on previous studies.
Response: The point behind choosing these variables was motivated by the initial results. The study was at first focusing on knowledge and attitude towards this novel virus and disease. But based on previous studies that pointed to the negative impact of belief in conspiracy regarding the origin of this pandemic, we tried to correlate such a belief to anxiety and knowledge which was illustrated in the results.
#3. I think that UJ students’ behavioral responses to COVID-19 (e.g., wearing masks) should be measured in the study. This is because it is very important for us to investigate how cognitive and emotional responses to COVID-19 may affect (or be related to) behavioral responses to COVID-19 in order to offer practical implications related to preventing the spread of COVID-19 in Jordan. I wondered what the authors think about my thoughts.
Response: Although we believe that such a question would be highly valuable, unfortunately we cannot answer it since the survey was closed.
#4. I believe that students at health schools are different from students at scientific schools in cognitively responding to COVID-19 (e.g., COVID-19 knowledge) as the virus is strongly associated with human health that is mainly studied by students at health schools but not students at scientific schools. Hence, I think that the school variable can be divided into three categories: health schools, scientific schools and humanities schools. I wondered what the authors think about my thoughts.
Response: Although we agree with the reviewer that students at Health Schools had more knowledge about the virus compared to Scientific Schools, we felt that grouping the two divisions together and comparing them to the students of the Humanities Schools would give better understanding of the results considering that Scientific Schools include students who study at the School of science with its affiliated divisions of Laboratory and biological sciences. Thus, we prefer to keep the analysis in its original form.
#5. I’d like to suggest that (1) discussion of ‘2.2. COVID-19 knowledge calculation’ sub-section is inserted after line 115 and (2) discussion of ‘2.3. Anxiety score calculation’ sub-sections is inserted after line 119, in order to increase readers’ understandability of measures.
Response: We prefer to keep the division as it is since these two parameters were key in our analysis and we feel that detailed description of each score would be more helpful for the readers.
Round 2
Reviewer 2 Report
I agree about the authors’ replies to my comments.
Because the sentence of “P-values less than 0.05 were considered significant.” (line 156) was described in the manuscript, the findings with p-values more than 0.05 needed to be deleted. For example, “Higher mean K-score was observed among those who felt that COVID-19 is very dangerous compared to those who felt that the disease is moderately dangerous, however, without statistical significance (p=0.150; K-W).” (lines 219-221) and “In addition, higher level of anxiety was also noticed among non-Jordanians, however, without statistical significance (mean anxiety score: 9.2 vs. 8.3, p=0.068; M-W).” (lines 242-244)
I wish you all the best with the manuscript!